# Assessing Fairness in the Presence of Missing Data

**Yiliang Zhang**
University of Pennsylvania
Philadelphia, PA 19104, USA
zylthu14@sas.upenn.edu

**Qi Long** *
University of Pennsylvania
Philadelphia, PA 19104, USA
qlong@upenn.edu

## Abstract

Missing data are prevalent and present daunting challenges in real data analysis. While there is a growing body of literature on fairness in analysis of fully observed data, there has been little theoretical work on investigating fairness in analysis of incomplete data. In practice, a popular analytical approach for dealing with missing data is to use only the set of complete cases, i.e., observations with all features fully observed to train a prediction algorithm. However, depending on the missing data mechanism, the distribution of complete cases and the distribution of the complete data may be substantially different. When the goal is to develop a fair algorithm in the complete data domain where there are no missing values, an algorithm that is fair in the complete case domain may show disproportionate bias towards some marginalized groups in the complete data domain. To fill this significant gap, we study the problem of estimating fairness in the complete data domain for an arbitrary model evaluated merely using complete cases. We provide upper and lower bounds on the fairness estimation error and conduct numerical experiments to assess our theoretical results. Our work provides the first known theoretical results on fairness guarantee in analysis of incomplete data.

## 1 Introduction

Mounting evidence [20, 8, 26, 42, 43] has suggested that powerful machine learning algorithms can be unfair and lead to disproportionately unfavorable treatment decisions for marginalized groups. In recent years, there has been a growing body of research on addressing the unfairness and bias of machine learning algorithms [12].

Meanwhile, missing data are ubiquitous and present daunting challenges in real-world data analysis. Particularly, missing data, if not adequately handled, would lead to biased estimations and improper statistical inferences [29]. As such, analysis of incomplete data has been an active research area [4, 33]. More recently, there is also a growing recognition that missing data may have deleterious impact on algorithmic fairness. For example, in medicine, bias caused by missing values in electronic health records is identified as a significant factor contributing to unfairness of machine learning (ML) algorithms used in medicine that may exacerbate health care disparities [36, 20]. However, there has been little reported research on assessing fairness of statistical and machine learning models using datasets that contain missing values.

In the presence of missing data, one popular approach for analysis, particularly in biomedical studies, is to use only the set of complete cases, i.e., observations with all features observed, discarding incomplete cases. As a result, we can define two related yet distinct data domains, namely, the complete case domain and the complete data domain (see Section 2.1). In many biomedical applications,

---

*This work is partly supported by NIH grant R01GM124111 and RF1AG063481. The content is the responsibility of the authors and does not necessarily represent the views of NIH.

35th Conference on Neural Information Processing Systems (NeurIPS 2021).

samples in the complete data domain are considered to be randomly drawn from the target population of interest; in other words, the complete data domain is defined by the target population, and the ultimate goal is to apply the trained models to the complete data domain. Under some missing data mechanisms (see Section 2.1), the complete case domain is biased for estimation of fairness in the complete data domain, i.e., for the target population. As such, a fair algorithm in the complete case domain may show disproportionate bias towards marginalized groups in the complete data domain.

The growing body of literature on algorithmic fairness has been primarily focused on two types of fairness definitions, group fairness and individual fairness [12]. Group fairness emphasizes that members from different groups (e.g. gender, race etc.) should be treated similarly, while individual fairness pays more attention to treatment similarity between any two similar individuals. In this work, we investigate group fairness in analysis of incomplete data. It has been noted that fairness definitions may not be compatible with one another in a sense that it is not possible to achieve fairness simultaneously under multiple definitions [19]. In binary classification problems, *demographic (or statistical) parity* [10, 17] is a fairness notion that has been mostly studied. It states that the predicted outcome should be independent of sensitive attributes. However demographic parity can cause severe harm to prediction performance when the response is dependent of sensitive attributes. As an alternative, *disparate mistreatment* [47] states that misclassification level (e.g., in terms of overall accuracy, false negative rate, false discovery rate) should be similar between two sensitive groups. Similarly, [24] proposed *equalized odds*, which requires both false positive rate (FPR) and false negative rate (FNR) to be the same between two groups. In the regression setting, fairness is usually associated with the parity of loss between two groups [1, 35, 16]. To fix ideas, we propose to use in this paper *accuracy parity gap* as the fairness notion in learning tasks including classification and regression. We consider the technique of re-weighting on assessing fairness of a given algorithm using complete cases, in which different complete cases are assigned different weights. We show that if the weights are properly chosen, such approach can mitigate the estimation bias induced by the difference of domains. It is worth noting that our results can be generalized to other fairness notions such as equal opportunity and prediction error parity with respect to mean square error.

Several existing works [18, 44, 48, 39, 30, 21] are related to our work, but there are a number of fundamental differences between these works and ours. [18, 44, 48] investigated the impact of missing data on the fairness of downstream prediction models. [39] investigated fairness across different domains and provided an upper bound of fairness in the target domain given fairness estimate in the source domain. But their work does not deal with missing data and associated challenges and does not consider the technique of re-weighting. In addition, they provided only upper bounds on transferring fairness. [30] empirically evaluated fairness in the presence of missing data. They did not observe consistent fairness results when comparing different methods for handling missing values such as imputation. We provide theoretical guarantees on assessing fairness of algorithms via analysis of incomplete data. [21] proposed a causal graph-based framework for modeling the data missingness to guide the design of fair algorithms. But they considered the case when the entire sample is missing, while we consider all three missing mechanisms.

**Our contributions:** This work offers four novel contributions. First, it provides new insights for algorithmic fairness in analysis of incomplete data, which has not yet been well-investigated from a theoretical perspective. Second, we characterize the role and impact of the missing data mechanism on correction for the data domain shift in fairness estimation through the analysis of incomplete data. Third, there has been limited theoretical work on domain adaptation problems when estimated weights are used to correct for domain shift. [13] considers the setting where true inverse probability weights are known, [41] considers the case when weights are estimated, but their analysis of algorithm's generalization performance is compared with a clipped empirical risk and does not provide any lower bound. In this work we investigate the setting where the weights are estimated from the data based on correctly or incorrectly specified propensity score models under a given missing data mechanism. Besides, both upper and lower bounds for fairness estimation bias are provided. Fourth, while the estimand that is of primary interest in domain adaptation literature is the prediction accuracy, the estimand of our interest takes the form of absolute difference between two prediction accuracy terms. To obtain a tighter upper bound, we conduct proper relaxations and incorporate Bennett's inequality in the proof of Theorem 1, which is more nuanced. In addition, novel proof techniques are also presented in the proof of Theorem 2 after applying Bernstein's inequalities, which does not appear in existing works for establishing lower bounds in domain adaptation problems such as [13].

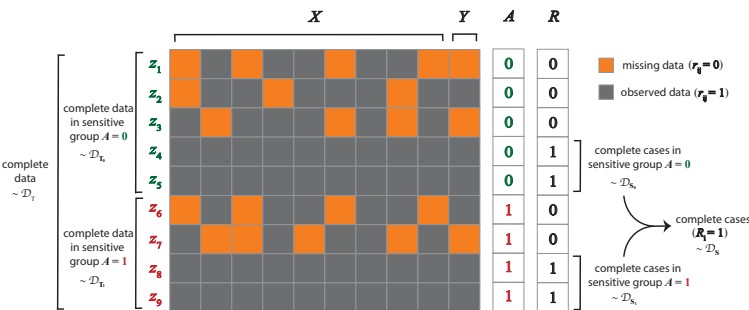

**Figure 1:** Missing data structure and notation. $\mathbf{Z}_{(1)}$ includes all observed data elements and $\mathbf{Z}_{(0)}$ includes all missing data elements. $R$ is the indicator for complete cases ($R = 1$) vs incomplete cases ($R = 0$). $A$ is the binary sensitive attribute.

## 2 Problem Formulation

### 2.1 Preliminaries on missing data

The data structure and notation used in this paper mostly follow the convention in the missing data literature [29] and are summarized in Figure 1. Suppose we have a random sample of $n$ observations from a target population of interest. If there were no missing values, each observation/case $\boldsymbol{z}_i :=$ $\{\boldsymbol{x}_i, y_i\} \in \mathcal{X} \times \mathcal{Y}$ ($i = 1, \ldots, n$) consists of predictors $\boldsymbol{x}_i \in \mathcal{X}$ and label (response variable) $y_i \in \mathcal{Y}$. Denote the complete data matrix by $\mathbf{Z}$ in which the $i^{\text{th}}$ row is denoted by $\boldsymbol{z}_i$. Some entries of $\mathbf{Z}$ are missing and we define the indicator for observing $z_{ij}$ or not as $r_{ij} = 1_{z_{ij} \text{ is observed}}$, where $z_{ij}$ is the $j$-th feature in $\boldsymbol{z}_i$. Denote the corresponding indicator matrix by $\mathbf{R}$. Let $\boldsymbol{z}_{(1)i}$ denote the components of $\boldsymbol{z}_i$ that are observed for observation $i$, and $\boldsymbol{z}_{(0)i}$ denote the components of $\boldsymbol{z}_i$ that are missing for observation $i$. For example, consider the case when there are two predictors and one response; if only $z_{i1}$ is observed, then $\boldsymbol{z}_{(1)i} = z_{i1}$, $\boldsymbol{z}_{(0)i} = (z_{i2}, z_{i3})$. We then define the observed data $\mathbf{Z}_{(1)}$ as the collection of the observed components from all $n$ observations, $\left\{\boldsymbol{z}_{(1)i}, i = 1, \ldots, n\right\}$ and the missing data $\mathbf{Z}_{(0)}$ as the collection of all the missing components, $\left\{\boldsymbol{z}_{(0)i}, i = 1, \ldots, n\right\}$.

There are three primary missing data mechanisms, namely, *missing completely at random* (MCAR), *missing at random* (MAR) and *missing not at random* (MNAR) [29]. Data are said to be MCAR if the distribution of $\mathbf{R}$ is independent of $\mathbf{Z}$. For MAR, the distribution of $\mathbf{R}$ depends on $\mathbf{Z}$ only through its observed components, i.e., $\mathbf{R} \perp \mathbf{Z}_{(0)} | \mathbf{Z}_{(1)}$. For MNAR, the distribution of $\mathbf{R}$ depends on the missing components of $\mathbf{Z}$. We seek to investigate fairness guarantee under all three mechanisms.

In the presence of missing data, observation $i$ is said to be a *complete case* if it is fully observed (i.e. $\boldsymbol{z}_{(1)i} = \boldsymbol{z}_i$). Let $R_i := \mathbf{1}_{\boldsymbol{z}_i \text{ is fully observed}}$ denote the indicator of complete cases. We can then define two different data domains (distributions) and we use the two terms, domain and distribution, interchangeably in the remainder of the paper. The complete data domain, denoted by $\mathcal{D}_T$, is the distribution of $\boldsymbol{z}_i$ in the joint space $\mathcal{X} \times \mathcal{Y}$. The complete case domain, denoted by $\mathcal{D}_S$, is the distribution of observations that have all variables fully observed: $\boldsymbol{z}_i | R_i = 1$. There is an important connection between these two domains and the aforementioned $\mathbf{Z}_{(1)}$ and $\mathbf{Z}_{(0)}$. $\mathbf{Z}_{(1)}$ contains the data in all the complete cases and the observed data in the incomplete cases. Combining $\mathbf{Z}_{(1)}$ and $\mathbf{Z}_{(0)}$ yields the complete data, which follows the distribution $\mathcal{D}_T$ (Figure 1).

Under MCAR, the distribution of the complete cases in $\mathcal{D}_S$ is the same as the distribution of the complete data in $\mathcal{D}_T$. However under MAR or MNAR, the data distributions in $\mathcal{D}_S$ and $\mathcal{D}_T$ can be different. For example, if missingness depends on gender and other features associated with gender, then females may have a substantially higher proportion of missing values and the feature distribution in females in $\mathcal{D}_S$ may be very different from that in $\mathcal{D}_T$. As a result, an algorithm that is trained to be fair in $\mathcal{D}_S$ may not be fair when evaluated in $\mathcal{D}_T$, noting again that in practice we typically are more interested in fairness guarantee in $\mathcal{D}_T$. We can view this as a domain shift from $\mathcal{D}_S$ to $\mathcal{D}_T$. This shift in the setting of incomplete data can be characterized by the true propensity score (a.k.a, the probability of being a complete case) [23]. To mitigate the fairness estimation bias caused by the shift, different weights can be assigned to the data in the fairness estimator. Missing data mechanism plays a vital role in obtaining the 'right' weight. For MCAR, uniform weights are equivalent to the

true weights. For MAR, one can train popular nonparametric models to obtain a consistent estimator for the true weights. For MNAR, it's much more difficult to obtain consistent estimators for the true weights. Hence one of our contributions is to characterize the role and impact of the missing data mechanism on bias correction in fairness estimation through the analysis of incomplete data.

## 2.2 Fairness estimand

We consider learning tasks that use features $x \in \mathcal{X}$ to predict response $y \in \mathcal{Y}$. Each observation also has a binary sensitive attribute $A \in \{0, 1\}$. We are interested in assessing fairness of a prediction model $g : \mathcal{X} \to \mathcal{Y}$ in the complete data domain $\mathcal{D}_T$. Let $\mathcal{E}_a(g) := \mathbb{E}_{T_a} |g(x) - y(x)|$ denote the prediction error, where $T_a$ represents that the expectation is taken with respect to $\mathcal{D}_{T_a}$, the distribution of complete cases that belongs to sensitive group $A = a$ (Figure 1). To derive our theoretical results, we consider the following fairness estimand.

**Definition 1** (Accuracy Parity Gap). For a given prediction model $g$, the *accuracy parity gap* of $g$ is $\Delta_T(g) = |\mathcal{E}_0(g) - \mathcal{E}_1(g)|$, where subscript $T$ indicates that the fairness estimand is defined in the complete data domain $\mathcal{D}_T$.

This definition has close connections with various fairness notions proposed in the literature. In binary classification tasks where the response is binary: $y \in \{0, 1\}$, the notion *accuracy parity* has been used in [47, 19, 50], which requires that the prediction accuracy between two sensitive groups to be equal: $P(g(x) = y|A = 0) = P(g(x) = y|A = 1)$. Accuracy parity gap in this case is the absolute value of difference between above two quantities $|P(g(x) = y|A = 0) - P(g(x) = y|A = 1)|$. For regression tasks where the response $y$ takes continuous value, fairness constraints on loss difference between two groups are adopted in [16, 35, 1]. Accuracy parity gap under such setting can be regarded as the difference of the mean absolute error (MAE) loss between two sensitive groups.

## 2.3 Fairness estimator

To estimate $\Delta_T(g)$, one can first estimate $\mathcal{E}_0(g)$ and $\mathcal{E}_1(g)$ using the set of complete cases in $\mathcal{D}_S$. However, the resulting estimator can be biased because of the difference between $\mathcal{D}_T$ and $\mathcal{D}_S$ under MAR and MNAR. To mitigate such estimation bias, one useful approach is to assign weight $\omega(z_i, A_i)$ to observation $z_i$ with sensitive attribute $A_i$, and calculate the weighted sum over the complete cases to estimate $\mathcal{E}_0(g)$ and $\mathcal{E}_1(g)$. Specifically, we define the weighted empirical risk (prediction error) using the complete cases as $\widehat{\mathcal{E}}_a(g, \omega) := \frac{1}{\sum_{i=1}^n I(A_i=a)R_i} \sum_{i=1}^n I(A_i = a)R_i\omega(z_i, A_i)|g(x_i) - y_i|$, where $a \in \{0, 1\}$. Here we assume there is at least one complete case observed for each sensitive group (i.e. $\sum_{i=1}^n I(A_i = a)R_i \geq 1$). Then the proposed fairness estimator is defined as follows.

**Definition 2** (Fairness estimator from complete cases). Suppose the weights assigned to complete cases are given by $\omega$. Then the fairness estimator for prediction model $g$ in the complete data domain is $\widehat{\Delta}_S(g, \omega) = |\widehat{\mathcal{E}}_0(g, \omega) - \widehat{\mathcal{E}}_1(g, \omega)|$, where subscript $S$ indicates that the estimator is obtained from the data from the complete case domain $\mathcal{D}_S$.

An ideal choice of $\omega$ is the normalized inverse of the propensity score, which can effectively mitigate the bias caused by the difference of $\mathcal{D}_T$ and $\mathcal{D}_S$. Mathematically, we let $\pi(z_i, A_i) := P_T(R_i = 1|z_i, A_i)$ denote the true propensity score (PS) model. In practice, we typically do not know the true propensity scores or the true distribution of complete cases, so we need to estimate the propensity scores and use the empirical distribution of complete cases. Various statistical and machine learning models can be used to estimate the propensity scores, such as logistic regression, random forest [7], support vector machines [15] and boosting algorithms [45, 32].

## 3 Main Results

In this section we provide the main theoretical results for the proposed fairness estimator $\widehat{\Delta}_S(g, \omega)$ in the complete data domain $\mathcal{D}_T$. Throughout, we assume that the weights are normalized in the complete case domain for both sensitive groups $a \in \{0, 1\}$: $\mathbb{E}_{S_a}\omega(z, a) = 1$ and are bounded away from 0 and infinity. Here $S_a$ represents that the expectation is taken with respect to $\mathcal{D}_{S_a}$, the distribution of complete cases in the sensitive group $A = a$. Of note, if the weights $\omega(z, a)$ are independent of $z$, then $\omega(z, a) \equiv 1$. It is commonly assumed in the domain adaptation literature that,

without loss of generality, $y$ in the regression setting takes value inside interval $[b_1, b_2]$ for some real numbers $b_1$ and $b_2$ [14, 49].

While our theoretical results are derived for APG in Definition 1, the framework of our proofs can be adapted to other fairness notions and the resulting forms of $\mathcal{E}_a(g)$. Take the example of a fairness notion for the regression problem, which is defined as the difference of $L^p$ loss with $1 \leq p < \infty$. The form of the covering number and the variance term $\text{Var}_{S_i}(\omega(\boldsymbol{z})|g(\boldsymbol{x}) - y(\boldsymbol{x})|)$ in the analysis should be adjusted. For binary classification, our framework can be adapted to all the five measurements of disparate mistreatment (accuracy, false positive rate, false negative rate, false mission rate and false discover rate) mentioned in Section 2 of [47]. Notably, *false negative rate parity* is also known as *Equal Opportunity* proposed in [24].

## 3.1  An upper bound

Let $B = \sup_{\boldsymbol{z}} \omega(\boldsymbol{z}, A) < +\infty$ denote the upper bound of the weights, $D_a^\omega = \mathbb{E}_{S_a} \omega(\boldsymbol{z}, a)^2$ denote the second moment of weights. $\omega \mathcal{D}_{S_a}$ denotes the distribution whose probability density function at $\boldsymbol{z}$ equals to $\omega(\boldsymbol{z}, a) f_{S_a}(\boldsymbol{z})$, with $f_{S_a}$ being the probability density function in $\mathcal{D}_{S_a}$. We further let $n_a$ denote the number of complete cases in group $a \in \{0, 1\}$ defined by the sensitivity feature. Without loss of generality, throughout the paper we assume $n_0 \leq n_1$, that is, sensitive group $A = 0$ is always the minority group. Theorem 1 below provides an upper bound of the fairness estimation bias.

**Theorem 1.** *Assume that $g$ is from a hypothesis class $\mathcal{H}$ with VC dimension $d$ (pseudo dimension if in the regression setting) and $y \in [0, 1]$. Assume $D_a^\omega \leq n_a/8$ for both groups. Then, for any $\delta > 0$, with probability at least $1 - \delta$, the following inequality holds:*

$$\left| \Delta_T(g) - \widehat{\Delta}_S(g, \omega) \right| \leq \sum_{a \in \{0,1\}} d_{TV}\left(\mathcal{D}_{T_a} \| \omega \mathcal{D}_{S_a}\right) + \sqrt{\frac{BC_d(n_a, D_a^\omega, \delta)}{n_a \left[ (1 + \frac{D_a^\omega}{B}) \log(1 + \frac{B}{D_a^\omega}) - 1 \right]}} \quad (1)$$

*where $d_{TV}\left(\mathcal{D}_{T_a} \| \omega \mathcal{D}_{S_a}\right)$ denote the total variation distance between $\mathcal{D}_{T_a}$ and $\omega \mathcal{D}_{S_a}$, and that*

$$C_d(n_a, D_a^\omega, \delta) = \begin{cases} \log \frac{(d+1)(8e)^{d+1}}{\delta} + \frac{d}{2} \log \frac{n_a}{2D_a^\omega} & \text{if } g \in \{0, 1\} \text{ is a classification model} \\ \log \frac{4}{\delta} \left( \frac{8e}{d} \right)^d + \frac{3d}{2} \log \frac{n_a}{\sqrt[3]{2D_a^\omega}} & \text{if } g \in [0, 1] \text{ is a regression model and } n_a \geq d \end{cases}$$

**Remark 1.** If $y \in [b_1, b_2]$, the upper bound in Theorem 1 would be multiplied by $b_2 - b_1$ under a new assumption $D_a^\omega(b_2 - b_1) \leq n_a/8$ for both sensitive groups.

**Remark 2.** $D_a^\omega$ is always upper bounded by $B^2$. In particular, for the MCAR mechanism, $D_a^\omega = 1$.

The detailed proof of Theorem 1 is in Appendix A. Since the fairness estimand is defined based on prediction error, there is some similarity between the upper bound in Theorem 1 and the upper bounds for learning error in the domain adaptation [6, 37] and survival analysis [5]. Our upper bound is obtained by the triangle inequality and the detailed analysis of generalization error. In the first term of the upper bound, $\omega \mathcal{D}_{S_a}$ is an approximation to the complete data domain in sensitive group $a$, and $d_{TV}(\mathcal{D}_{T_a} \| \omega \mathcal{D}_{S_a})$ can be viewed as the approximation error. It follows that a less accurate approximation of the complete data domain would lead to a looser upper bound on fairness estimation error. In the second term $C_d(n_a, D_a^\omega, \delta)$ is proportional to $\log n_a$, so the term is of order $\mathcal{O}(\log n_0/n_0)^{\frac{1}{2}}$. It also follows that for a fixed total number of complete cases, the upper bound increases with sample imbalance between two groups defined by $A$. In addition, the missing data mechanism impacts the second moment of estimated weights $D_a^\omega \in [1, B^2]$. If a missing data mechanism leads to a larger second moment of the weights, the upper bound for the fairness estimation error would be looser. Furthermore, when the weights $\omega$ are defined using the true propensity scores, we call the resulting $\omega_0(\boldsymbol{z}_i, A_i) = [\pi(\boldsymbol{z}_i, A_i) \mathbb{E}_S \{1/\pi(\boldsymbol{z}, A)|A = A_i\}]^{-1}$ as the true weights. When true weights are adopted, the first term in the upper bound in Theorem 1 vanishes and we have the following result.

**Corollary 1.** *If $\omega(\boldsymbol{z}_i, A_i) = \omega_0(\boldsymbol{z}_i, A_i)$, then $\widehat{\Delta}_S(g, \omega)$ is consistent for estimating $\Delta_T(g)$.*

There are several implications from Corollary 1. Under MCAR, setting $\omega(\boldsymbol{z}_i, A_i) = 1$ would yield a consistent (unweighted) estimator. Under MAR and MNAR, since we typically do not know the true propensity scores $\pi(\boldsymbol{z}_i, A_i)$, we replace $\pi(\boldsymbol{z}_i, A_i)$ with its estimate $\hat{\pi}(\boldsymbol{z}_i, A_i)$ using a working model

which is subject to mis-specification. If a correctly-specified propensity score model is adopted, the second term in the upper bound would be the dominant term, in which the upper bound can be approximated from the observed data more easily. Otherwise, the first term would often be the dominant term, in which $d_{\mathrm{TV}}(\mathcal{D}_{T_a} \| \omega \mathcal{D}_{S_a})$ can be even larger than that of the unweighted estimator.

## 3.2 A lower bound

We define $\sigma_a^2(g, \omega) := \mathrm{Var}_{S_a}(\omega|g - y|)$. Given $\omega$ is upper bounded by $B$ and $g, y \in [0, 1]$, we have that $\sigma_a^2(g, \omega) \leq B^2/4$ by Popoviciu's inequality on variance. We present the result on a lower bound for the fairness estimation error in Theorem 2.

**Theorem 2.** *If the weight $\omega(\boldsymbol{z}_i, A_i)$ is set to be $\omega_0(\boldsymbol{z}_i, A_i)$ (true weights) and $B^2/\sigma_a^2(g, \omega) \leq n_0$, then the following hold with probability at least $\frac{7}{1440}$,*

$$12\sqrt{\frac{\sigma_0^2(g, \omega_0)}{n_0} + \frac{\sigma_1^2(g, \omega_0)}{n_1}} \geq \left|\left(\mathcal{E}_0(g) - \mathcal{E}_1(g)\right) - \left(\widehat{\mathcal{E}}_0(g, \omega) - \widehat{\mathcal{E}}_1(g, \omega)\right)\right| \geq \frac{1}{24}\sqrt{\frac{\sigma_0^2(g, \omega_0)}{n_0} + \frac{\sigma_1^2(g, \omega_0)}{n_1}}$$

(2)

*Additionally, if $\widehat{\Delta}_S(g, \omega) \geq \frac{13}{2}\sqrt{\frac{\sigma_0^2(g, \omega_0)}{n_0} + \frac{\sigma_1^2(g, \omega_0)}{n_1}}$, we have*

$$\left|\Delta_T(g) - \widehat{\Delta}_S(g, \omega)\right| \geq \frac{1}{24}\sqrt{\frac{\sigma_0^2(g, \omega_0)}{n_0} + \frac{\sigma_1^2(g, \omega_0)}{n_1}}$$

*If $\widehat{\Delta}_S(g, \omega) \leq \frac{1}{72}\sqrt{\frac{\sigma_0^2(g, \omega_0)}{n_0} + \frac{\sigma_1^2(g, \omega_0)}{n_1}}$, we have:*

$$\left|\Delta_T(g) - \widehat{\Delta}_S(g, \omega)\right| \geq \frac{1}{72}\sqrt{\frac{\sigma_0^2(g, \omega_0)}{n_0} + \frac{\sigma_1^2(g, \omega_0)}{n_1}}$$

The detailed proof of Theorem 2 is in Appendix B. The proof involves detailed analysis of the truncation probability for the fairness difference.

**Remark 3.** If $y$ is bounded in $[b_1, b_2]$ in regression instead of the unit interval, Theorem 2 still holds.

**Remark 4.** If the weight $\omega$ is based on the estimated $\hat{\pi}(\boldsymbol{z}_i, A_i)$ from a correctly specified propensity score model, then the optimal convergence rate for $|\omega(\boldsymbol{z}_i, A_i) - \omega_0(\boldsymbol{z}_i, A_i)|$ is $\mathcal{O}_p((n_0 + n_1)^{-1/2})$. Based on the results from (1) and (2), the upper bound is of order $\mathcal{O}((\log n_0/n_0)^{1/2})$. With some additional assumptions on the data distribution, the lower bound can be shown to have order $\mathcal{O}((n_0)^{-1/2})$. (See Appendix B for a more detailed analysis)

**Remark 5.** The bounds in Theorem 2 are established under fairly weak conditions on data distributions. If one is willing to make additional assumptions on the tail behavior (e.g., gaussian or sub-gaussian), the bounds would hold with higher probabilities.

**Remark 6.** There is very limited work on the lower bound analysis in the fields of domain adaptation and fairness. The existing works [13, 27] have reported similar results as ours, i.e., the lower bound of generalization error holds with a low probability.

Our theoretical results have additional important implications related to the missing data mechanisms. First, as long as sample size is sufficiently large as required, Theorem 1 would always hold for all mechanisms. Under the MCAR mechanism, the true propensity score is a constant and hence can be regarded as known, and the results from Theorem 2 hold for the unweighted estimator. Under the MAR mechanism, the true propensity score is generally unknown. Under the MNAR mechanism, the propensity score model depends on missing values, so it cannot be estimated without making additional modeling assumptions. If the propensity score model is mis-specified under MAR or MNAR, the results in Theorem 2 are not applicable.

## 4 Numerical Experiments

In this section we empirically evaluate the bias of fairness estimation in both synthetic and real data sets, particularly the results in Theorem 1 and Theorem 2. As a reminder, our goal is to estimate fairness $\Delta_T(g)$ (i.e., APG) defined in the complete data domain, while the estimator $\hat{\Delta}_S(g, \omega)$ is obtained from the complete case domain.

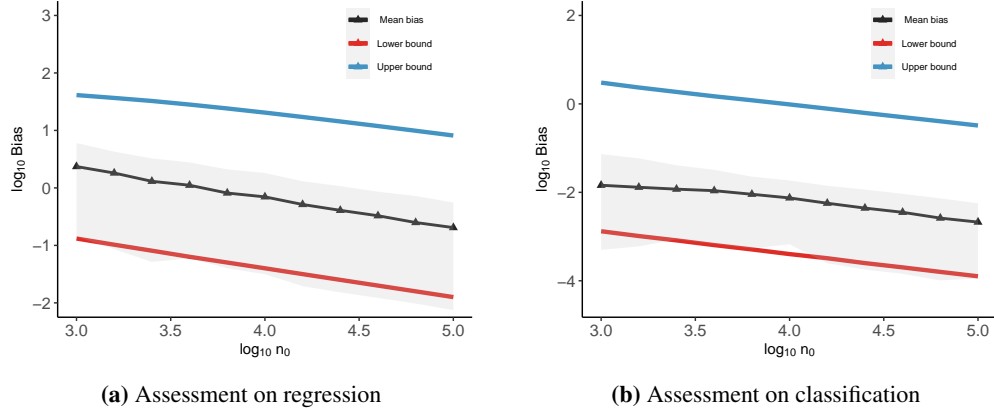

**(a)** Assessment on regression           **(b)** Assessment on classification

**Figure 2:** Synthetic data experiments on assessment of upper and lower bounds. Bias, $|\Delta_T(g) - \widehat{\Delta}_S(g, \omega)|$. The shaded area is the 90% confidence band for Bias, defined by 5th and 95th percentiles in the 500 repeated experiments.

## 4.1 Synthetic experiments

In our simulation experiments, we assess the upper bound in Theorem 1 in a classification task and the lower bound in Theorem 2 in a regression task. We further investigate the effects of different factors on the fairness estimation in regression tasks. In each experiment, we generate 10 predictors and a binary sensitive attribute $A \in \{0, 1\}$ with $n$ samples. Unless noted otherwise, the predictors are generated from Gaussian distributions: $x_{ij} \sim \mathcal{N}(1 - 2A_i, 0.5^2)$ ($i = 1, \ldots, n$ and $j = 1, \ldots, 10$). We generate missing values among the last five predictors, from a pre-specified propensity score model. The propensity score model and responses $y$ are set differently in different experiments. In all the experiments except the first one, we use a set of 2000 data (predictors are drawn from aforementioned distribution) to train a prediction algorithm $g$, where the dataset is balanced regarding the sensitive attribute. We use another set of data, which contains missing values, to calculate $\widehat{\Delta}_S(g, \omega)$. In our numerical experiments, it is difficult to obtain the analytical form of the true accuracy parity gap $\Delta_T(g)$, so we approximate $\Delta_T(g)$ using the Monte Carlo method with additional 100000 samples that are not used in obtaining $\widehat{\Delta}_S(g, \omega)$ or $g$.

**Assessment of the upper bound in Theorem 1:** In this experiment, we seek to answer the question: What practical guarantees can Theorem 1 provide for estimation of $\Delta_T(g)$, the true fairness of a prediction model $g$ in the complete data domain. Given $\hat{\Delta}_S(g, \omega)$ and the right hand side of (1), we can construct an interval in which, with a high probability, $\Delta_T(g)$ lies. We consider a binary classification problem with response $y_i \sim \text{Bernoulli}((1 + \exp(\boldsymbol{x}_i^\top \beta))^{-1})$, where $\beta = (0.1, 0.1, 0.1, 0.1, 0.1, 1, 1, 1, 1, 1)^\top$. Each sensitive group contains 50000 samples and the missingness of data is drawn according to the following propensity score model $\text{logit}(\pi(\boldsymbol{z}_i, A_i)) = 0.25 - 0.5A_i$, where $\text{logit}(p) = \log \frac{p}{1-p}$. We use the complete cases among aforementioned samples to build a linear support vector machine $g$. Since the propensity score model only depends on sensitive attribute $a$, by definition, the true weights $\omega_0(\boldsymbol{z}, a) \equiv 1$. We calculate $\hat{\Delta}_S(g, \omega)$ from the complete cases with $\omega = 1$ and calculate the right hand side of (1). It follows from Theorem 1 that with probability at least 95%, the true fairness $\Delta_T(g)$ is covered by $[0, 0.25]$. We use additional 100000 data to approximate the APG value $\Delta_T(g) \approx 0.13$. In this experiment setting, we observe that the upper bound in (1) is not significantly loose and might be able to provide useful information about the fairness of $g$ in practice (e.g., $g$ is not extremely unfair in the complete data domain).

**Assessment of the lower bound in Theorem 2:** In this experiment, we assess the upper and lower bounds on both regression and classification problems. Let $\beta = (0.1, 0.1, 0.1, 0.1, 0.1, 1, 1, 1, 1, 1)^\top$ and $\epsilon = (\epsilon_1, \ldots, \epsilon_n)^\top \sim \mathcal{N}(0, I_n)$. For regression problem, we consider $y_i = (\boldsymbol{x}_i^\top \beta)^2 + \epsilon_i$, for classification problem, we consider $y_i \sim \text{Bernoulli}((1 + \exp(\boldsymbol{x}_i^\top \beta))^{-1})$. We use linear SVM as the prediction model $g$, and the missingness of data is drawn from the following propensity score model $\text{logit}(\pi(\boldsymbol{z}_i, A_i)) = -3 + \frac{1}{5}\sum_{j=1}^{5} x_{ij}$. Of note, since $R_i$ depends on only the fully observed features, the missing data mechanism is MAR. In this experiment, we use the true weights $\omega(\boldsymbol{z}, A) = \omega_0(\boldsymbol{z}, A)$ to calculate the fairness estimation bias $|\Delta_T(g) - \hat{\Delta}_S(g, \omega)|$. We check the assumptions and plot both

bounds in Figure 2. Observe that the lower bound from Theorem 2 is always smaller than the mean of fairness estimation bias, as the sample size $n_0$ in the minority sensitive group changes from $10^2$ to $10^4$ while the ratio of $n_0/n$ is fixed at 1/2. Besides, in both problems, the lower bounds are close to the 5th percentiles of the fairness estimation bias, implying that they are not disproportionately loose. Our results from this and other unreported experiments lend support to that for certain data distributions such as those in this experiment, the lower bound may hold with a considerably higher probability than stated in Theorem 2. In addition, the slope of (average) fairness estimation bias is approximately the same as that of the lower bound, indicating the convergence rate of fairness estimation bias to be $\mathcal{O}(n_0^{-1/2})$ in this experiment setting. In addition, we notice that the confidence bands in Figure 2 seem to have equal length, given different sample sizes. However, since the $y$-axis has a log scale, the actual length of confidence band shrinks with increasing sample size.

**Impact of different weights:** We consider the same regression task in previous experiment, except that the predictors $x$ are drawn from $\mathcal{N}(3 - 6A_i, 0.5^2)$. Missingness of data is drawn under MAR using the following propensity score model $\text{logit}(\pi(\boldsymbol{z}_i, A_i)) = -\frac{1}{5}\sum_{j=1}^{5} x_{ij}$. We fix the sample ratio between two sensitive groups in the complete cases, which guarantees that both groups have the same number of complete cases (in expectation). In all the remaining synthetic experiments, we adopt random forest as the prediction model $g$. We calculate $\hat{\Delta}_S(g, \omega)$ using seven different weights, namely, $\omega(\boldsymbol{z}_i, A_i) = 1$ (i.e., unweighted estimator), the true weights $\omega(\boldsymbol{z}_i, A_i) = \omega_0(\boldsymbol{z}_i, A_i)$ and 5 weights obtained via $\omega(\boldsymbol{z}_i, A_i) = \hat{\omega}_0(\boldsymbol{z}_i, A_i) := \frac{R_i}{\hat{\pi}(\boldsymbol{z}_i, A_i)\sum_{j=1}^{n}\{I(A_j=A_i)R_j/\hat{\pi}(\boldsymbol{z}_j, A_j)\}}$ where $\hat{\pi}(\boldsymbol{z}_i, A_i)$ is estimated from two different logistic regression models (correctly and incorrectly specified), random forest (RF) [7], support vector machine (SVM) [15] and extreme gradient boosting (XGB) [11]. In particular we use $\{x_{ij}\}$ to fit the first logistic regression model, which is correctly specified. In the second model, we use $\{x_{ij}^3\}$, which leads to an incorrectly specified logistic model. Figure 3-(a) shows that using the true weights or correctly specified logistic regression model yields similar performance and leads to smaller fairness estimation bias $|\Delta_T(g) - \hat{\Delta}_S(g, \omega)|$ than the incorrectly-specified logistic model and the other propensity score models. At the first glance, it might seem surprising that XGB and random forest, as more expressive algorithms, do not outperform other propensity score estimators. A potential reason is that the true propensity score is the probability of observing all variables in a sample, which is often a number between 0 and 1 (bounded away from 0 by assumption). However, the data used to fit a propensity score model are binary labels (0 for missing data and 1 for observed data). The more expressive models such as XGB and random forest is more likely to predict label 1 for observed data and 0 for missing data accurately, but it does not always yield more accurate estimation of the true propensity score.

**Impact of sample imbalance:** We consider the same regression task defined in the previous experiment. We vary the total sample size $n$ from $10^3$ to $10^5$, and for each fixed $n$ we examine different levels of sample imbalance between the two sensitive groups by varying the ratio of $n_1/n_0$ from 1 to 20. We draw missingness of data under MAR from the following propensity score model, $\text{logit}(\pi(\boldsymbol{z}_i, A_i)) = -1 + \frac{1}{5}\sum_{j=1}^{5} x_{ij}$. To estimate the fairness $\Delta_T(g)$, we use estimated propensity scores from the correctly specified logistic regression model. The resulting fairness estimation bias $|\Delta_T(g) - \hat{\Delta}_S(g, \omega)|$ is shown in Figure 3-(b). For a fixed $n$ increasing sample imbalance leads to larger bias, suggesting that sample imbalance could harm the fairness estimation.

**Impact of disparity in data distribution between sensitive groups:** We consider the same regression task with $n$ varying from $10^3$ to $10^5$, in which $x$ is drawn from $\mathcal{N}(1 - 2MA_i, 0.5^2)$ with $M$ controlling the difference in data distribution between the two sensitive groups. Missingness of data is generated under MAR using the following model $\text{logit}(\pi(\boldsymbol{z}_i, A_i)) = 2 - 4A_i$. We use a correctly-specified logistic regression model to obtain the weights for estimating $\hat{\Delta}_S(g, \omega)$. As shown in Figure 3-(c), the fairness estimation bias increases as $M$ increases for a given $n$. This suggests that it can be harder to guarantee fairness in the complete data domain when the disparity in data distribution between the two sensitive groups become more pronounced.

## 4.2   Real data experiments

We conduct analyses of two real datasets, one from the COMPAS and the other from the ADNI. We consider the task of building prediction model and assess its fairness (in the target population) using the same dataset containing missing values. In each experiment, we randomly split the real dataset

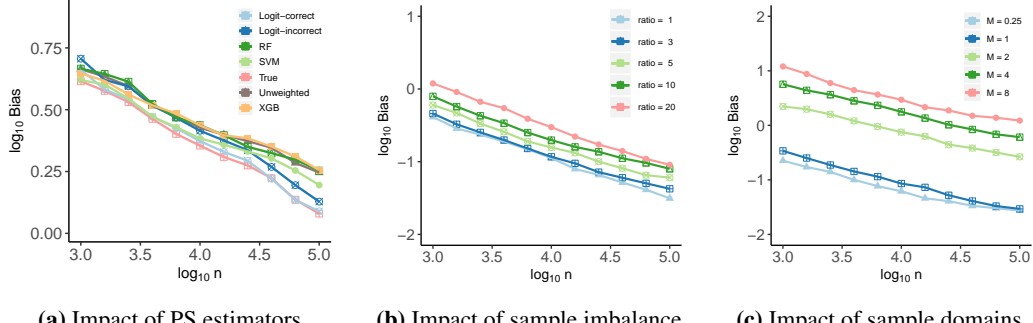

| | (a) Impact of PS estimators | (b) Impact of sample imbalance | (c) Impact of sample domains |

**Figure 3:** Synthetic data experiments on effects of different factors on the fairness estimation. Bias, $|\Delta_T(g) - \widehat{\Delta}_S(g,\omega)|$. In (b), ratio is $n_1/n_0$. In (c), $M$ controls the disparity between data distributions from the two sensitive groups.

into two subsets. In the first subset, we generate missing values, and the complete cases in this subset are used to train a random forest prediction model $g$ and estimate its fairness in the complete data domain. The true fairness $\Delta_T(g)$ is approximated using the entire second subset. Particularly, we generate missing values under three settings, namely, MCAR, MAR and MNAR respectively.

| | | | | Specification of $\omega$ | | | |
|---|---|---|---|---|---|---|---|
| | | Unweighted | True | Logistic | RF | SVM | XGB |
| COMPAS | MCAR | **1.16 ± 0.96** | 1.14 ± 0.88 | 1.17 ± 0.97 | **1.16 ± 0.90** | 1.37 ± 1.27 | 1.19 ± 0.95 |
| $(\times 10^{-2})$ | MAR | 1.52 ± 1.09 | 1.14 ± 0.89 | **1.24 ± 0.99** | 1.46 ± 1.01 | 46.5 ± 31.2 | 1.37 ± 1.17 |
| | MNAR | 5.32 ± 2.31 | 3.83 ± 2.12 | 5.11 ± 2.03 | 5.32 ± 2.22 | 9.64 ± 7.18 | **5.05 ± 2.20** |
| ADNI | MCAR | 2.90 ± 2.73 | 2.98 ± 2.75 | 3.05 ± 2.72 | 2.95 ± 2.79 | **2.86 ± 2.78** | 2.95 ± 2.76 |
| $(\times 10^{-3})$ | MAR | 3.80 ± 3.67 | 3.66 ± 3.46 | 4.61 ± 3.96 | 7.35 ± 6.00 | **3.79 ± 3.42** | 3.90 ± 3.39 |
| | MNAR | **3.80 ± 3.45** | 3.58 ± 3.07 | 3.86 ± 3.31 | 6.16 ± 5.47 | 3.88 ± 3.27 | 3.84 ± 3.31 |

**Table 1:** Bias in fairness estimation $|\Delta_T(g) - \widehat{\Delta}_S(g,\omega)|$ with different options for $\omega$ and missing data mechanisms in analysis of the COMPAS and ADNI datasets. Mean ± SD over 100 repeats.

To assess the impact of weight specifications, we compare multiple options of $\omega$, a) $\omega(\boldsymbol{z}_i, A_i) = 1$ (i.e., unweighted), b) the true inverse probability weights $\omega(\boldsymbol{z}_i, A_i) = \omega_0(\boldsymbol{z}_i, A_i)$, and c) $\omega(\boldsymbol{z}_i) = \widehat{\omega}_0(\boldsymbol{z}_i, A_i) = R_i/(\hat{\pi}(\boldsymbol{z}_i, A_i) \sum_{j=1}^n \{I(A_j = A_i)R_j/\hat{\pi}(\boldsymbol{z}_j, A_j)\})$ where $\hat{\pi}(\boldsymbol{z}_i, A_i)$ is obtained from logistic regression, RF, SVM and XGB. To evaluate the impact of sample imbalance between two sensitive groups, we fix the total sample size and use logistic regression to compute weights $\widehat{\omega}_0(\boldsymbol{z}_i, A_i)$. In all real data analyses, the logistic regression model is the correctly specified model for $\pi(\boldsymbol{z}_i, A_i)$ under MAR mechanisms. Of note, while we can obtain fairness estimators (in the complete case domain) using real datasets with actual missing values, we would not be able to assess the bias in fairness estimation since the true fairness of an algorithm (in the complete data domain) would not be available. In addition, we would not be able to use the true propensity score model for missingness as a valuable benchmark. As such, we chose to generate artificial missing values in real datasets, which allows us to compute bias in estimating fairness and use the true propensity score model as a benchmark while making the experiments more realistic.

**COMPAS recidivism dataset:** Correctional Offender Management Profiling for Alternative Sanctions (COMPAS) [34] is a risk assessment instrument developed by Northpointe Inc. The dataset analyzed in this work contains records of defendants from Broward County from 2013 and 2014. Prior work has demonstrated the bias of predictions from COMPAS towards certain groups of defendants defined by race, gender and age etc. [2]. In our analysis, gender is treated as the sensitive attribute and nine numerical features are used to predict two-year recidivism (defined by arrest within 2 years) [38]. We generate missing values for the last feature and the outcome variable under three missing mechanisms: MCAR, $\text{logit}(\pi(\boldsymbol{z}_i, A_i)) = 0.8$; MAR, $\text{logit}(\pi(\boldsymbol{z}_i, A_i)) = 3 + 2\sum_{j=1}^5 x_{ij}$; MNAR, $\text{logit}(\pi(\boldsymbol{z}_i, A_i)) = -2y - 2x_{i9}$. As shown in Table 1, all options of $\omega$ lead to comparable results under MCAR, noting that all of them including the unweighted estimator are valid under MCAR.

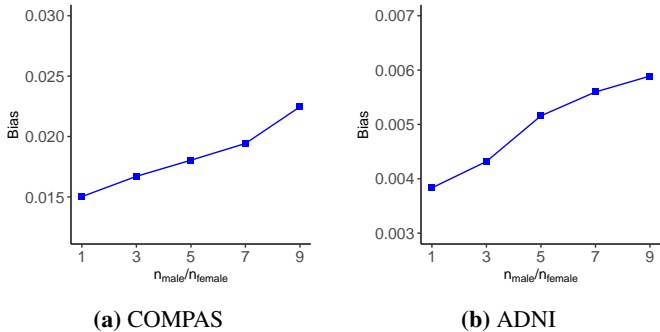

**(a)** COMPAS           **(b)** ADNI

**Figure 4:** Impact of sample imbalance in two real datasets from 500 repeated experiments. Mean fairness estimation biases are plotted.

Under MAR, the use of true weights leads to the least bias and followed by logistic regression, noting that logistic regression is the correctly specified model for $\pi(z_i, A_i)$ in this data analysis. Under MNAR, all working propensity score models are mis-specified and yield larger bias than the true weights. To study the impact of sample imbalance, we fix the total sample size as 800 and vary the proportion of samples in the two sensitive groups, $n_{\text{male}}/n_{\text{female}}$, from 1 to 9. The curve of fairness estimation bias is shown in Figure 4. The results show that in more imbalanced data, fairness estimation bias is larger, consistent with our findings in the simulations.

**ADNI gene expression data:** The dataset from the Alzheimer's Disease Neuroimaging Initiative (ADNI) contains gene expression and clinical data for 649 patients. In our analysis, training set contains 500 samples and we only include the top 1000 transcriptomic features with highest positive correlation with gender, the sensitive feature. The outcome variable is the VBM right hippocampal volume. Missing values are generated for the last 900 features under the three missing data mechanisms: MCAR, $\text{logit}(\pi(z_i, A_i)) = 0.5$; MAR, $\text{logit}(\pi(z_i, A_i)) = -2 - \frac{1}{5}\sum_{j=1}^{10} x_{ij}$; MNAR, $\text{logit}(\pi(z_i, A_i)) = -2 - \frac{1}{5}\sum_{j=101}^{110} x_{ij}$. As shown in Table 1, the main findings are consistent with those from the analysis of the COMPAS dataset. Regarding the effect of sample imbalance with a fixed total sample size 150, Figure 4 displays the curve of fairness estimation bias, showing the same patterns as in the analysis of the COMPAS dataset.

## 5 Discussions

This work provides the first known theoretical results on fairness guarantee in analysis of incomplete data. The bounds in Theorems 1 and 2 quantify the convergence rates of fairness estimation, which can help understand the impact of the sample size and guide choosing adequate sample size in practice. In addition, if the weights used in fairness estimation are not close to the true propensity score weights, the total variation distance term in the upper bound may become the dominant term. Thus, it is vital to use proper weights in fairness estimation. Our result can be adapted to other domain shift settings such as in the presence of selection bias where certain groups are under-represented due to biased sampling. A limitation of our work is that we only consider the analysis of complete cases. Another commonly-used alternative approach is to impute missing values first and then assess the fairness of the prediction model. However, there are several potential challenges for assessing fairness guarantee after imputation. Fairness estimation based on imputed data depends on the specific imputation method used, as different imputation methods have different operating characteristics and theoretical properties. In particular, the distributional properties of imputed data would play a vital role in understanding the theoretical properties of fairness estimation using imputed data. Such properties remain under-investigated for many popular imputation methods. For example, theoretical properties of multiple imputation via chain equations [9] and missForest [40] have not been well-established except for some very restrictive settings. Existing works on matrix completion (e.g. [31]) are focused on assessing imputation error, and there is little work on their statistical properties. Theoretical properties of popular deep learning imputation methods such as Mida [22], Gain [46] and misGan [28] have not been studied. We anticipate our work to lay a foundation for future research on assessing fairness in the presence of missing data.

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
