# A  Proof of Theorem 1

We begin by stating and proving the following theorem, which will play an important role in the proof of Theorem 1. Suppose we have $N$ observations $\{z_i\}_{i=1}^N$ drawn from an arbitrary domain $\mathcal{D}_S$ and let

$$\widehat{\mathcal{E}}(g,\omega) := \frac{1}{\sum_{i=1}^N R_i} \sum_{i=1}^N R_i \omega(z_i, A_i) |g(x_i) - y_i|$$

Furthermore we define $D^\omega = \mathbb{E}_S \omega(z, A)^2$, in this section we use $\mathbb{E}_S$ to represent that the expectation is taken with respect to this arbitrary domain $\mathcal{D}_S$. Then we have the following result:

**Theorem 3.** *Let $\mathcal{H}$ be a hypothesis set with VC-dimension (pseudo dimension in regression setting) $d$. Let $g$ be an arbitrary prediction model from hypothesis set $\mathcal{H}$. If $D^\omega \leq N/8$, then, for any $\delta > 0$, with probability at least $1 - \delta$, the following bound holds:*

$$\left| \mathbb{E}_S \omega(z, A) |g(x) - y(x)| - \widehat{\mathcal{E}}(g, \omega) \right| \leq \sqrt{\frac{BC_d(N, D^\omega, \delta)}{N \left[ (1 + \frac{D^\omega}{B}) \log(1 + \frac{B}{D^\omega}) - 1 \right]}} \tag{3}$$

$$C_d(N, D^\omega, \delta) = \begin{cases} \log \frac{(d+1)(8e)^{d+1}}{2\delta} + \frac{d}{2} \log \frac{N}{2D^\omega} & \text{if } g \in \{0, 1\} \text{ is a classification model} \\ \log \frac{2}{\delta} \left( \frac{8e}{d} \right)^d + \frac{3d}{2} \log \frac{N}{\sqrt[3]{2D^\omega}} & \text{if } g \in [0, 1] \text{ is a regression model and } N \geq d \end{cases}$$

*Proof.* The proof follows a standard approach in deriving generalization error bound related to VC-dimension (or pseudo-dimension). Recall that an observation $z = \{x, y\}$, we begin by letting $f_g(z) := \omega(z) |g(x) - y(x)|$. Then since $g \in \mathcal{H}$, we let $\mathcal{F}$ denote the set of $f_g$. In the rest of the proof, we simply ignore subscription $g$ and let $f(z) := \omega(z) |g(x) - y(x)|$. This is possible since the analysis holds for arbitrary $g \in \mathcal{H}$, i.e. $f \in \mathcal{F}$. We simplify the notation by defining

$$\widehat{\mathcal{E}}(g, \omega) = \mathbb{P}_N f(z)$$

and

$$\mathbb{E}_S \omega(z) |g(x) - y(x)| = \mathbb{P} f(z)$$

We further let $\mathbb{D}_N$ denote the training data $\{(x_i, y_i)\}_{i=1}^N$. We also consider a set of "ghost" sample $\mathbb{D}'_N = \{(x'_i, y'_i)\}_{i=1}^N$ with size $N$. In reality we do not have access to them but we will make use of them to prove the theorem. We let $f_N$ be the maximizer of $|\mathbb{P}f - \mathbb{P}_N f|$ in $\mathcal{F}$ and $\sigma^2 = \text{Var}[f_N]$. Similar to $\mathbb{P}_N$, we can define $\mathbb{P}'_N$ for the ghost samples. Firstly notice that

$$I\left(|\mathbb{P}f_N - \mathbb{P}_N f_N| > t\right) I\left(|\mathbb{P}f_N - \mathbb{P}'_N f_N| < t/2\right) \leq I\left(|\mathbb{P}'_N f_N - \mathbb{P}_N f_N| > t/2\right)$$

Taking expecation with respect to the ghost sample yields

$$I\left(|\mathbb{P}f_N - \mathbb{P}_N f_N| > t\right) P_{D'_N}\left(|\mathbb{P}f_N - \mathbb{P}'_N f_N| < t/2\right) \leq P_{\mathbb{D}'_N}\left(|\mathbb{P}'_N f_N - \mathbb{P}_N f_N| > t/2\right)$$

Chebyshev's inequality gives

$$P_{D'_N}\left(|\mathbb{P}f_N - \mathbb{P}'_N f_N| > t/2\right) \leq \frac{4\text{Var}[f_N]}{Nt^2} = \frac{4\sigma^2}{Nt^2} \leq \frac{4D^\omega}{Nt^2}$$

where the last inequality is given by the fact that $D^\omega = \mathbb{E}\omega^2 \geq \text{Var}[f_N] = \sigma^2$ (see Lemma 2 in [13]). This in turn gives

$$I\left(|\mathbb{P}f_N - \mathbb{P}_N f_N| > t\right)\left(1 - \frac{4D^\omega}{Nt^2}\right) \leq P_{D'_N}\left(|\mathbb{P}'_N f_N - \mathbb{P}_N f_N| > t/2\right)$$

when $t \geq \sqrt{\frac{8D^\omega}{N}}$, we have

$$P\left(\sup_{f \in \mathcal{F}} |\mathbb{P}f - \mathbb{P}_N f| \geq t\right) \leq 2P\left(\sup_{f \in \mathcal{F}} |\mathbb{P}'_N f - \mathbb{P}_N f| \geq t/2\right) \tag{4}$$

We further define $\mathcal{F}_{|N} = \{(f(x_1, y_1), \ldots, f(x_N, y_N)) \mid f \in \mathcal{H}\}$. Then

$$P\left(\sup_{f \in \mathcal{F}} |\mathbb{P}'_N f - \mathbb{P}_N f| \geq t/2\right) = P\left(\sup_{f \in \mathcal{F}_{|2N}} |\mathbb{P}'_N f - \mathbb{P}_N f| \geq t/2\right)$$

Let $\mathcal{N}_1(t/8, \mathcal{F}, 2N)$ denote the uniform covering number defined as

$$\mathcal{N}_1(t/8, \mathcal{F}, 2N) = \max_{\mathbb{D}_N, \mathbb{D}'_N} \mathcal{N}\left(t/8, \mathcal{F}_{|2N}, d_1\right) \tag{5}$$

where $\mathcal{N}\left(t/8, \mathcal{F}_{|2N}, d_1\right)$ is the $t/8$-covering number of set $\mathcal{F}_{|2N}$ with respect to $L^1$ distance. Now define $\mathcal{G} \subseteq \mathcal{F}_{|2N}$ as a $t/8$-cover of $\mathcal{F}_{|2N}$ with $|\mathcal{G}| \leq \mathcal{N}_1(t/8, \mathcal{F}, 2N)$. Then

$$P\left(\sup_{f \in \mathcal{F}_{|2N}} |\mathbb{P}'_N f - \mathbb{P}_N f| \geq t/2\right) \leq P\left(\sup_{g \in \mathcal{G}} |\mathbb{P}'_N g - \mathbb{P}_N g| \geq t/4\right)$$

$$\leq \mathcal{N}_1(t/8, \mathcal{F}, 2N) \sup_{g \in \mathcal{G}} P\left(|\mathbb{P}'_N g - \mathbb{P}_N g| \geq t/4\right) \tag{6}$$

$$\leq 2\mathcal{N}_1(t/8, \mathcal{F}, 2N) \sup_{g \in \mathcal{G}} P\left(|\mathbb{P}_N g - \mathbb{P}g| \geq t/8\right)$$

Let us first consider the case of classification, in which the predicted outcome $g(x) \in \{0, 1\}$. Then according to classical bound of covering number (Theorem 1 in [25]):

$$\mathcal{N}\left(t/8, \mathcal{F}_{|2N}, d_1\right) < e(d+1)\left(\frac{16e}{t}\right)^d$$

The part remained is analysis of the probability $\sup_{g \in \mathcal{G}} P\left(|\mathbb{P}_N g - \mathbb{P}g| \geq t/8\right)$. Bennett's inequality gives:

$$\sup_{g \in \mathcal{G}} P\left(|\mathbb{P}_N g - \mathbb{P}g| \geq t/8\right) \leq \exp\left(\frac{-N\sigma^2 h(Bt/\sigma^2)}{B^2}\right) \tag{7}$$

with $h(u) := (1+u)\log(1+u) - u$. Notice that here $t \leq 1$, we further have that $h_{B,\sigma}(t) = h(Bt/\sigma^2)$ is lower bounded by function

$$\underline{h}(t) = \left[(1 + \frac{B}{\sigma^2})\log(1 + \frac{B}{\sigma^2}) - \frac{B}{\sigma^2}\right] t^2$$

The argument is obtained by observing that $\underline{h}(0) = h_{B,\sigma}(0)$, $\underline{h}(1) = h_{B,\sigma}(1)$, $\underline{h}(0)' = h_{B,\sigma}(0)'$ and that $\underline{h}(0)''$ is decreasing while $h_{B,\sigma}(0)''$ is a constant. Together with equation (7), we have that

$$\sup_{g \in \mathcal{G}} P\left(|\mathbb{P}_N g - \mathbb{P}g| \geq t/8\right) \leq \exp\left(-\frac{N}{B}\left[(1 + \frac{\sigma^2}{B})\log(1 + \frac{B}{\sigma^2}) - 1\right] t^2\right) \tag{8}$$

Combining the bound of covering number yields:

$$P\left(\sup_{f \in \mathcal{F}} |\mathbb{P}f - \mathbb{P}_N f| \geq t\right) < 4e(d+1)\left(\frac{16e}{t}\right)^d \exp\left(-\frac{N}{B}\left[(1 + \frac{\sigma^2}{B})\log(1 + \frac{B}{\sigma^2}) - 1\right] t^2\right)$$

$$\leq 4e(d+1)\left(16e\sqrt{\frac{N}{8D^\omega}}\right)^d \exp\left(-\frac{N}{B}\left[(1 + \frac{\sigma^2}{B})\log(1 + \frac{B}{\sigma^2}) - 1\right] t^2\right) \tag{9}$$

Let $\delta = 4e(d+1)\left(16e\sqrt{\frac{N}{8D^\omega}}\right)^d \exp\left(-\frac{N}{B}\left[(1 + \frac{\sigma^2}{B})\log(1 + \frac{B}{\sigma^2}) - 1\right] t^2\right)$. Simplify the equation gives

$$\frac{N}{B}\left[(1 + \frac{\sigma^2}{B})\log(1 + \frac{B}{\sigma^2}) - 1\right] t^2 - C_d(N, D^\omega, \delta) = 0$$

where

$$C_d(N, D^\omega, \delta) = \log \frac{(d+1)(8e)^{d+1}}{2\delta} + \frac{d}{2}\log \frac{N}{2D^\omega}$$

This equation has non-negative solution

$$t_\delta = \sqrt{\frac{BC_d(N, D^\omega, \delta)}{N\left[(1 + \frac{\sigma^2}{B})\log(1 + \frac{B}{\sigma^2}) - 1\right]}}$$

Thus we have

$$1 - \delta \le P\left(\sup_{f \in \mathcal{F}} |\mathbb{P}f - \mathbb{P}_N f| \le t_\delta\right) \le P\left(\sup_{f \in \mathcal{F}} |\mathbb{P}f - \mathbb{P}_N f| \le \sqrt{\frac{BC_d(N, D^\omega, \delta)}{N\left[(1 + \frac{\sigma^2}{B})\log(1 + \frac{B}{\sigma^2}) - 1\right]}}\right)$$

Now we consider the regression case, in which the predicted outcome $g(\boldsymbol{x}) \in [0, 1]$ comes from the hypothesis class $\mathcal{H}$ with pseudo dimension $d$. Notice that all the results above hold before equation (6). Now by Theorem 12.2 in [3], we have that

$$\mathcal{N}\left(t/8, \mathcal{F}_{|2N}, d_1\right) \le \mathcal{N}\left(t/8, \mathcal{F}_{|2N}, d_\infty\right) \le \left(\frac{16Ne}{td}\right)^d$$

when $N \ge d/2$. Combining with (7) yields

$$P\left(\sup_{f \in \mathcal{F}} |\mathbb{P}f - \mathbb{P}_N f| \ge t\right) < 2\left(\frac{16Ne}{td}\right)^d \exp\left(-\frac{N}{B}\left[(1 + \frac{\sigma^2}{B})\log(1 + \frac{B}{\sigma^2}) - 1\right]t^2\right)$$

$$\le 2\left(\frac{16Ne}{d}\sqrt{\frac{N}{8D^\omega}}\right)^d \exp\left(-\frac{N}{B}\left[(1 + \frac{\sigma^2}{B})\log(1 + \frac{B}{\sigma^2}) - 1\right]t^2\right)$$

Let $\delta = 2\left(\frac{16Ne}{d}\sqrt{\frac{N}{8D^\omega}}\right)^d \exp\left(-\frac{N}{B}\left[(1 + \frac{\sigma^2}{B})\log(1 + \frac{B}{\sigma^2}) - 1\right]t^2\right)$ and define

$$C_d(N, D^\omega, \delta) = \log\frac{2}{\delta}\left(\frac{8e}{d}\right)^d + \frac{3d}{2}\log\frac{N}{\sqrt[3]{2D^\omega}}$$

Following exactly the same procedure as above, we can have

$$1 - \delta \le P\left(\sup_{f \in \mathcal{F}} |\mathbb{P}f - \mathbb{P}_N f| \le t_\delta\right) \le P\left(\sup_{f \in \mathcal{F}} |\mathbb{P}f - \mathbb{P}_N f| \le \sqrt{\frac{BC_d(N, D^\omega, \delta)}{N\left[(1 + \frac{\sigma^2}{B})\log(1 + \frac{B}{\sigma^2}) - 1\right]}}\right)$$

Finally, notice that $D^\omega \ge \sigma^2$ and that $(1 + x)\log(1 + 1/x)$ is monotonic decreasing, we have that

$$P\left(\sup_{f \in \mathcal{F}} |\mathbb{P}f - \mathbb{P}_N f| \le \sqrt{\frac{BC_d(N, D^\omega, \delta)}{N\left[(1 + \frac{D^\omega}{B})\log(1 + \frac{B}{D^\omega}) - 1\right]}}\right) \ge 1 - \delta$$

for both classification and regression cases.

$\square$

Now we are ready to prove Theorem 1:

*Proof.* Notice that for both groups $a \in \{0, 1\}$:

$$\left|\mathcal{E}_a(g) - \mathbb{E}_{S_a}\omega(\boldsymbol{z}, a)|g(\boldsymbol{x}) - y(\boldsymbol{x})|\right| = \left|\mathbb{E}_{S_a}(\omega_0(\boldsymbol{z}, a) - \omega(\boldsymbol{z}, a))|g(\boldsymbol{x}) - y(\boldsymbol{x})|\right| \quad (10)$$

By triangle inequality, combining (10) in Theorem 3 and (3) yields that with at least probability $1 - 2\delta$:

$$\left|\left|\mathcal{E}_0(g) - \mathcal{E}_1(g)\right| - \left|\widehat{\mathcal{E}}_0(g, \omega) - \widehat{\mathcal{E}}_1(g, \omega)\right|\right|$$

$$\le \sum_{a \in \{0, 1\}} \left|\mathbb{E}_{S_a}(\omega_0(\boldsymbol{z}, a) - \omega(\boldsymbol{z}, a))|g(\boldsymbol{x}) - y(\boldsymbol{x})|\right| + \left|\mathbb{E}_{S_a}\omega(\boldsymbol{z}, a)|g(\boldsymbol{x}) - y(\boldsymbol{x})| - \widehat{\mathcal{E}}_a(g, \omega)\right|$$

$$\le \sum_{a \in \{0, 1\}} \left|\mathbb{E}_{S_a}(\omega_0(\boldsymbol{z}, a) - \omega(\boldsymbol{z}, a))|g(\boldsymbol{x}) - y(\boldsymbol{x})|\right| + \sqrt{\frac{BC_d(n_a, D_a^\omega, \delta)}{n_a\left[(1 + \frac{D_a^\omega}{B})\log(1 + \frac{B}{D_a^\omega}) - 1\right]}}$$

$$\quad (11)$$

Notice that by definition of total variation distance:

$$d_{\text{TV}}(\mathcal{D}_{T_a} || \omega \mathcal{D}_{S_a}) \ge \left|\mathbb{E}_{S_a}(\omega_0(\boldsymbol{z}, a) - \omega(\boldsymbol{z}, a))|g(\boldsymbol{x}) - y(\boldsymbol{x})|\right|$$

Finally substituting $\delta$ to $\delta/2$ yields the result.

$\square$

# B  Proof of Theorem 2

*Proof.* Since we have $\omega = \omega_0$, we only use $\omega$ in the proof for the sake of simplicity. The proof is inspired by the technique used in Theorem 9 in [13]. Assume $(\boldsymbol{x}, y) = \{(\boldsymbol{x}_i, y_i)\}_{i=1}^{n_0+n_1}$ is the complete cases, in which $n_0$ of the data belongs to sensitive group $A = 0$. Let

$$
\phi_g(\boldsymbol{x}, y) = \widehat{\mathcal{E}}_0(g, \omega) - \widehat{\mathcal{E}}_1(g, \omega)
$$
$$
= \frac{1}{n_0} \sum_{i=1}^{n_0} \omega((\boldsymbol{x}_{0,i}, y_{0,i}), 0) |g(\boldsymbol{x}_{0,i}) - y_{0,i}| - \frac{1}{n_1} \sum_{i=1}^{n_1} \omega((\boldsymbol{x}_{1,i}, y_{1,i}), 1) |g(\boldsymbol{x}_{1,i}) - y_{1,i}|
$$

with $g \in \mathcal{H}$. Obviously when true weights are adopted, we have $\mathbb{E}\phi_g(\boldsymbol{x}, y) = \mathcal{E}_0(g) - \mathcal{E}_1(g)$. Without loss of generality, we assume $n_0 < n_1$. We let $\sigma_i^2 := \mathrm{Var}_{S_i}(\omega|g - y|)$. Furthermore let $\sigma^2 = \sigma_0^2 + (n_0/n_1)\sigma_1^2$. Now consider $U := \frac{\mathbb{E}\phi_g(\boldsymbol{x}, y) - \phi_g(\boldsymbol{x}, y)}{\sigma}$. Notice that

$$
\mathbb{E}Z^2 = \frac{\frac{1}{n_0}\sigma_0^2 + \frac{1}{n_1}\sigma_1^2}{\sigma^2} = \frac{1}{n_0} \tag{12}
$$

Meanwhile we can split the expectation into

$$
\mathbb{E}U^2 \mathbf{1}_{|U| \in [0, 1/(k\sqrt{n_0}))} + \mathbb{E}U^2 \mathbf{1}_{|U| \in [1/(k\sqrt{n_0}), u/\sqrt{n_0})} + \mathbb{E}U^2 \mathbf{1}_{|U| \in [u/\sqrt{n_0}, +\infty)}
$$

which is upper bounded by

$$
\frac{1}{k^2 n_0} + \frac{u^2}{n_0} P(u/\sqrt{n_0} > |U| > 1/(k\sqrt{n_0})) + \mathbb{E}U^2 \mathbf{1}_{|U| \in [u/\sqrt{n_0}, +\infty)}
$$

Combined with (12) yields

$$
P(u/\sqrt{n_0} > |U| > 1/(k\sqrt{n_0})) \geq \frac{k^2 - 1}{k^2 u^2} - \frac{n_0}{u^2} \mathbb{E}U^2 \mathbf{1}_{|U| \in [u/\sqrt{n_0}, +\infty)} \tag{13}
$$

Now notice that $n_0 \mathbb{E}U^2 \mathbf{1}_{|U| \in [u/\sqrt{n_0}, +\infty)}$ can be written as

$$
n_0 \mathbb{E}U^2 \mathbf{1}_{|U| \in [u/\sqrt{n_0}, +\infty)} = \int_0^{+\infty} P\left[n_0|U|^2 \mathbf{1}_{|U| > \frac{u}{\sqrt{n_0}}} > t\right] dt
$$
$$
= \int_0^{u^2} P\left[|U| > \frac{u}{\sqrt{n_0}}\right] dt + \int_{u^2}^{+\infty} P\left[|U| > \sqrt{\frac{t}{n_0}}\right] dt \tag{14}
$$
$$
= u^2 P\left[|U| > \frac{u}{\sqrt{n_0}}\right] + \int_{u^2}^{+\infty} P\left[|U| > \sqrt{\frac{t}{n_0}}\right] dt
$$

The probability in the last line can be upper bounded by

$$
P\left[|U| > \sqrt{\frac{t}{n_0}}\right] \leq P\left[|(\frac{1}{n_0}\sum_{i=1}^{n_0} \omega(\boldsymbol{x}_{0,i}, y_{0,i})|g(\boldsymbol{x}_{0,i}) - y_{0,i}| - \mathbb{E}_{S_0}\omega(\boldsymbol{z}, 0)|g(\boldsymbol{x}) - y|)| > \frac{\sigma}{2}\sqrt{\frac{t}{n_0}}\right]
$$
$$
+ P\left[|(\frac{1}{n_1}\sum_{i=1}^{n_1} \omega(\boldsymbol{x}_{1,i}, y_{1,i})|g(\boldsymbol{x}_{1,i}) - y_{1,i}| - \mathbb{E}_{S_1}\omega(\boldsymbol{z}, 1)|g(\boldsymbol{x}) - y|)| > \frac{\sigma}{2}\sqrt{\frac{t}{n_0}}\right]
$$
$$
\leq \exp\left(-\frac{\sigma^2 t}{8\sigma_0^2 + 4/3 B\sigma\sqrt{\frac{t}{n_0}}}\right) + \exp\left(-\frac{(n_1/n_0)\sigma^2 t}{8\sigma_1^2 + 4/3 B\sigma\sqrt{\frac{t}{n_0}}}\right)
$$

where the second inequality is given by Bernstein's inequality. We now state that

$$
\frac{(n_1/n_0)\sigma^2 t}{8\sigma_1^2 + 4/3 B\sigma\sqrt{\frac{t}{n_0}}} \geq \frac{t}{8 + 4/3\sqrt{t}}
$$

To see this, consider two cases $\sigma > \sigma_1$ and $\sigma \leq \sigma_1$. If $\sigma > \sigma_1$ then

$$
\frac{(n_1/n_0)\sigma^2 t}{8\sigma_1^2 + 4/3 B\sigma\sqrt{\frac{t}{n_0}}} \geq \frac{\sigma^2 t}{8\sigma^2 + 4/3 B\sigma\sqrt{\frac{t}{n_0}}} \geq \frac{t}{8 + 4/3\sqrt{t}}
$$

where the second inequality is given by the assumption $B^2/\sigma^2 \leq n_0$. If $\sigma \leq \sigma_1$

$$\frac{(n_1/n_0)\sigma^2 t}{8\sigma_1^2 + 4/3B\sigma\sqrt{\frac{t}{n_0}}} = \frac{t((n_1/n_0)\sigma_0^2 + \sigma_1^2)}{8\sigma_1^2 + 4/3B\sigma\sqrt{\frac{t}{n_0}}} \geq \frac{t\sigma_1^2}{8\sigma_1^2 + 4/3B\sigma_1^2\sqrt{\frac{t}{n_0}}} \geq \frac{t}{8 + 4/3\sqrt{t}}$$

Similarly $\frac{\sigma^2 t}{8\sigma_0^2 + 4/3B\sigma\sqrt{\frac{t}{n_0}}} \geq \frac{t}{8+4/3\sqrt{t}}$. Notice that $\sqrt{t} \geq 1/k$, take $k = 24$, we have

$$P\left[|U| > \sqrt{\frac{t}{n_0}}\right] \leq 2\exp\left(-\frac{t}{8 + 4/3\sqrt{t}}\right) \leq 2\exp\left(-\frac{3\sqrt{t}}{5}\right) \tag{15}$$

Plug into (14) gives that

$$n_0\mathbb{E}U^2\mathbf{1}_{|U|\in[u/\sqrt{n_0},+\infty)} \leq 2u^2\exp\left(-\frac{3u}{5}\right) + \int_{u^2}^{+\infty} 2\exp\left(-\frac{3\sqrt{t}}{5}\right) dt$$

$$= 2\left(u^2 + \frac{10u}{3} + 2\left(\frac{5}{3}\right)^2\right)\exp\left(-\frac{3u}{5}\right)$$

when $u = 12$, above is smaller than $0.29$. In this case, (13) yields

$$P(|U| > 1/(24\sqrt{n_0})) > P(u/\sqrt{n_0} > |U| > 1/(24\sqrt{n_0})) = \frac{575}{576u^2} - \frac{0.29}{u^2} = \frac{7}{10u^2} = \frac{7}{10}\left(\frac{1}{12}\right)^2 \tag{16}$$

Finally we have the observation

$$P(|U| > 1/(24\sqrt{n_0})) = P\left[\left|\left(\mathcal{E}_0(g) - \mathcal{E}_1(g)\right) - \left(\widehat{\mathcal{E}}_0(g,\omega) - \widehat{\mathcal{E}}_1(g,\omega)\right)\right| > \frac{1}{24}\sqrt{\frac{\sigma_0^2}{n_0} + \frac{\sigma_1^2}{n_1}}\right]$$

which completes the proof of first argument. To see the second argument, it can be proven that when

$$\widehat{\Delta}_S(g,\omega) \geq \frac{13}{2}\sqrt{\frac{\sigma_0^2(g,\omega)}{n_0} + \frac{\sigma_1^2(g,\omega)}{n_1}} > \frac{u + 1/24}{2}\sqrt{\frac{\sigma_0^2(g,\omega)}{n_0} + \frac{\sigma_1^2(g,\omega)}{n_1}}$$

the interval $\left[\widehat{\Delta}_S(g,\omega) - u/\sqrt{n_0}, \widehat{\Delta}_S(g,\omega) + u/\sqrt{n_0}\right]$ will never intersect with the interval $\left[-\widehat{\Delta}_S(g,\omega) - 1/(24\sqrt{n_0}), -\widehat{\Delta}_S(g,\omega) + 1/(24\sqrt{n_0})\right]$. Hence we have

$$P\left[\left|\Delta_T(g) - \widehat{\Delta}_S(g,\omega)\right| > \frac{1}{24}\sqrt{\frac{\sigma_0^2(g,\omega)}{n_0} + \frac{\sigma_1^2(g,\omega)}{n_1}}\right] > P(u/\sqrt{n_0} > |U| > 1/(24\sqrt{n_0}))$$

The third argument can be proved in a similar way: when $\widehat{\Delta}_S(g,\omega) \leq \frac{1}{72}\sqrt{\frac{\sigma_0^2(g,\omega)}{n_0} + \frac{\sigma_1^2(g,\omega)}{n_1}}$, we have that

$$\left|\Delta_T(g) - \widehat{\Delta}_S(g,\omega)\right| \geq \left|\left(\mathcal{E}_0(g) - \mathcal{E}_1(g)\right) - \left(\widehat{\mathcal{E}}_0(g,\omega) - \widehat{\mathcal{E}}_1(g,\omega)\right)\right| - 2\widehat{\Delta}_S(g,\omega)$$

$$\geq \frac{1}{72}\sqrt{\frac{\sigma_0^2(g,\omega)}{n_0} + \frac{\sigma_1^2(g,\omega)}{n_1}}$$

$\square$

At the end of this section, we would like to provide a brief discussion about the case when the true weights $\omega_0$ are estimated by $\omega$, from a correctly-specified propensity score model. In this case, we apply Theorem 2 to $w$ and obtain that:

$$\left|\left(\mathbb{E}_{S_0}\omega(\boldsymbol{x}, 0)|g(\boldsymbol{x}) - y(\boldsymbol{x})| - \mathbb{E}_{S_1}\omega(\boldsymbol{x}, 1)|g(\boldsymbol{x}) - y(\boldsymbol{x})|\right) - \left(\widehat{\mathcal{E}}_0(g,\omega) - \widehat{\mathcal{E}}_1(g,\omega)\right)\right|$$

$$\geq \frac{1}{24}\sqrt{\frac{\sigma_0^2(g,\omega)}{n_0} + \frac{\sigma_1^2(g,\omega)}{n_1}} \tag{17}$$

with probability at least $\frac{7}{10}\left(\frac{1}{12}\right)^2$. We make two additional assumptions:

- The propensity score model is correctly specified and satisfies: $|\omega - \omega_0| = \mathcal{O}_p((n_0 + n_1)^{-1/2})$

- The prediction model satisfies that for arbitrary $\boldsymbol{x}$, $\mathbb{E}|g(\boldsymbol{x}) - y(\boldsymbol{x})| \xrightarrow{P} 0$ where the expectation is taken with respect to $y$ given $\boldsymbol{x}$.

From above we have that $D_a^{\omega} - D_a^{\omega_0} = \mathbb{E}_{S_a}\omega^2 - \mathbb{E}_{S_a}\omega_0^2 = \mathcal{O}_p((n_0 + n_1)^{-1/2})$ for both groups $a \in \{0, 1\}$. This yields

$$\sqrt{\frac{\sigma_0^2(g, \omega)}{n_0} + \frac{\sigma_1^2(g, \omega)}{n_1}} = \sqrt{\frac{\sigma_0^2(g, \omega_0)}{n_0} + \frac{\sigma_1^2(g, \omega_0)}{n_1}} + o_p(n_0^{-1/2} + n_1^{-1/2}) \qquad (18)$$

Meanwhile, we have

$$\left| \mathbb{E}_{S_a}\omega(\boldsymbol{x}, a)|g(\boldsymbol{x}) - y(\boldsymbol{x})| - \mathcal{E}_a(g) \right| = \left| \mathbb{E}_{S_a}(\omega - \omega_0)|g(\boldsymbol{x}) - y(\boldsymbol{x})| \right| = o_p((n_0 + n_1)^{-1/2}) \quad (19)$$

for both groups. Therefore,

$$\left| \left( \mathcal{E}_0(g) - \mathcal{E}_1(g) \right) - \left( \widehat{\mathcal{E}}_0(g, \omega) - \widehat{\mathcal{E}}_1(g, \omega) \right) \right|$$

$$= \left| \left( \mathbb{E}_{S_0}\omega_0(\boldsymbol{x}, 0)|g(\boldsymbol{x}) - y(\boldsymbol{x})| - \mathbb{E}_{S_1}\omega_0(\boldsymbol{x}, 1)|g(\boldsymbol{x}) - y(\boldsymbol{x})| \right) - \left( \widehat{\mathcal{E}}_0(g, \omega) - \widehat{\mathcal{E}}_1(g, \omega) \right) \right|$$

$$\geq \left| \left( \mathbb{E}_{S_0}(\omega(\boldsymbol{x}, 0) - \omega_0(\boldsymbol{x}, 0))|g(\boldsymbol{x}) - y(\boldsymbol{x})| - \mathbb{E}_{S_1}(\omega(\boldsymbol{x}, 1) - \omega_0(\boldsymbol{x}, 1))|g(\boldsymbol{x}) - y(\boldsymbol{x})| \right) \right|$$

$$- \left| \left( \mathbb{E}_{S_0}\omega(\boldsymbol{x}, 0)|g(\boldsymbol{x}) - y(\boldsymbol{x})| - \mathbb{E}_{S_1}\omega(\boldsymbol{x}, 1)|g(\boldsymbol{x}) - y(\boldsymbol{x})| \right) - \left( \mathcal{E}_0(g) - \mathcal{E}_1(g) \right) \right|$$

$$\geq \left| \left( \mathbb{E}_{S_0}(\omega(\boldsymbol{x}, 0) - \omega_0(\boldsymbol{x}, 0))|g(\boldsymbol{x}) - y(\boldsymbol{x})| - \mathbb{E}_{S_1}(\omega(\boldsymbol{x}, 1) - \omega_0(\boldsymbol{x}, 1))|g(\boldsymbol{x}) - y(\boldsymbol{x})| \right) \right|$$

$$+ \frac{1}{24}\sqrt{\frac{\sigma_0^2(g, \omega)}{n_0} + \frac{\sigma_1^2(g, \omega)}{n_1}}$$

where the last inequality holds with probability at least $\frac{7}{1440}$. Combining equation (18) and (19), we know that for arbitrary $\delta$, there exists $N_0$ and $N_1$ such that whenever $n_0 > N_0$ and $n_1 > N_1$, with probability at least $1 - \delta$,

$$\left| \left( \mathbb{E}_{S_0}(\omega(\boldsymbol{x}, 0) - \omega_0(\boldsymbol{x}, 0))|g(\boldsymbol{x}) - y(\boldsymbol{x})| - \mathbb{E}_{S_1}(\omega(\boldsymbol{x}, 1) - \omega_0(\boldsymbol{x}, 1))|g(\boldsymbol{x}) - y(\boldsymbol{x})| \right) \right|$$

$$\leq \frac{1}{24}\sqrt{\frac{\sigma_0^2(g, \omega)}{n_0} + \frac{\sigma_1^2(g, \omega)}{n_1}} - \frac{1}{25}\sqrt{\frac{\sigma_0^2(g, \omega_0)}{n_0} + \frac{\sigma_1^2(g, \omega_0)}{n_1}}$$

Thus with probability at least $\frac{7}{1440} - \delta$,

$$\left| \left( \mathcal{E}_0(g) - \mathcal{E}_1(g) \right) - \left( \widehat{\mathcal{E}}_0(g, \omega) - \widehat{\mathcal{E}}_1(g, \omega) \right) \right| \geq \frac{1}{25}\sqrt{\frac{\sigma_0^2(g, \omega_0)}{n_0} + \frac{\sigma_1^2(g, \omega_0)}{n_1}}$$

Following the procedure as that of Theorem 2, we can prove that the lower bound have order $\mathcal{O}((n_0)^{-1/2})$. Of course, the bound also holds with a low probability.

## C   Experiment details

The experiments are run using R (version 3.6.1) on a single 6-Core Intel Core i7 (2.6GHz). We use R package 'e1071' (licensed under GPL-2 | GPL-3) for SVM, 'ranger' (used for prediction model, licensed under GPL-3) and 'randomForest' (used for PS model, licensed under GPL-2 | GPL-3) for random forest and 'xgboost' (licensed under Apache License (== 2.0)) for XGB. The 'randomForest' is a commonly-used package for random forest and 'ranger' provides an efficient implementation for the algorithm, which can be adapted to the learning task in high dimensions. (e.g., the prediction model in the real data experiments).

## Data Availability

The de-identified COMPAS dataset is publicly available at https://github.com/propublica/compas-analysis/blob/master/compas-scores-two-years.csv. The de-identified ADNI dataset is publicly available at http://adni.loni.usc.edu/.