# OpenReview forum: "Assessing Fairness in the Presence of Missing Data"
_NeurIPS.cc/2021/Conference — NeurIPS 2021 Poster_

### Official Review · Reviewer_ykYW · 2021-07-14

**Rating:** 6
**Confidence:** 3

**Summary:**

The paper presents  theoretical results about fairness and missing data. The authors show theoretical bounds for the fairness metric "accuracy parity gap" considering weights. The weights are estimated from the data based on correctly or incorrectly specified propensity score models under a given missing data mechanism.

Some experiments on synthetic and real data (where some parameters are controlled) are useful to demonstrate the validity of  the included theoretical.

**Limitations And Societal Impact:**

No negative impact.

**Main Review:**

The paper is easy to read and follow. The authors motivate properly the impact  of unfair machine learning models. They also introduce the common problem of having missing data in training instances. Some works have studied the combination of missing data with fairness issues, however not significant theoretical results related to fairness metrics have been developed. In this aspect, the authors are able to propose a  framework  where they formalise the problem and the analysed   metric  ("accuracy parity gap" ), as well as three missing data contexts: missing completely at random (MCAR), missing at random (MAR) and missing not at random (MNAR). In this framework, they include a weight w used to express the fairness difference between D_t (complete data domain) and D_s (complete data domain). Considering this estimated value w, the authors develop two results about the limits in APG. This is the main result of the paper.

After that, the authors include a set of synthetic experiments where they study the bias considering different values of w and different imbalance factors. The experiments show the Assessment of the upper and lower  bound in classification and regression. Additionally, they add experiments with real data, however in a controlled environment since  they generate missing values under the three settings, namely, MCAR, MAR and MNAR.

weak points of the paper:

* I do not see clearly the applicability of the proposed results. Although the author comment some ideas, I miss a clear example where they can show the benefits of the study.

* In this sense, I also miss experiments with real data with actual missing values.  Some studies have shown that there exist some patterns with missing data and fairness. The application of the proposed framework over one real case could be useful.

* Finally, it could be also interesting in see how the proposed result could be employed to define imputation methods that could mitigate the fairness inequalities.

**Time Spent Reviewing:**

3

---

> ### Author Response · Authors · 2021-08-10
> **Response to Reviewer 4**
>
> Thank you for your very thoughtful and constructive comments! We would like to provide point-by-point responses to your comments below:
>
> **R4-C1: Applicability of the proposed results**
>
>  As indicated in our response to Reviewer 3’s comment 2 (R3-C2), our theoretical results can provide valuable insights on the convergence rate of the fairness estimator and factors that may impact fairness estimation in practice as stated in the Discussion section. Under MCAR, the first tem in the upper bound would vanish so the upper bound can be estimated. When the sample size is large under MAR, the second term in the upper bound would dominate the first term and can be estimated.  We will include this discussion in the updated version, provided that additional space is allowed.
>
> **R4-C2: Experiments with real data with actual missing values**
>
> While we can obtain fairness estimators (in the complete case domain) on real datasets with actual missing values, we would not be able to assess bias in fairness estimation since the true fairness of an algorithm (in the complete data domain) would not be available. In addition, we would not be able to use the true propensity score model for missingness as a valuable benchmark. As a result, we chose to generate artificial missing values in real datasets, which allows us to compute and compare bias in estimating fairness and use the true propensity score model as a benchmark while making the experiments more realistic.
>
> **R4-C3: Refine imputation methods that could mitigate bias in fairness estimation**
>
> This is an excellent point and we believe this will be an exciting problem for future research. In Section 5, we discussed some challenges in this area. We have ongoing work in this area.

---

### Official Review · Reviewer_6mRT · 2021-07-15

**Rating:** 5
**Confidence:** 3

**Summary:**

This paper provides theoretical guarantees (upper and lower bounds) on the estimation error of fairness, in the setting where missing data is present and only the complete rows are used for estimation. In particular, the authors consider the “accuracy parity gap” as the fairness metric, which can be applied to both classification and regression tasks. The paper also includes some numerical experiments to validate the theoretical findings.

**Limitations And Societal Impact:**

Yes

**Main Review:**

Theoretical analysis of fairness estimation with incomplete data is largely missing in the algorithmic fairness literature and can be a valuable contribution. The results also apply to both classification and regression. The authors claim that the results can be generalized to other group fairness notions, but this claim is not supported with details.

While the theorems for upper and lower bounds can be useful in understanding the behavior of fairness estimation under different settings, I am not sure if they will have much practical impact. To obtain intervals of fairness guarantees using the theorems, it requires quantities that are not available in practical settings such as the total variation distance with the target distribution. Also, the lower bound only holds with a very low probability.

The authors also empirically study the effects of multiple factors on fairness estimation which is a nice contribution, but the results are somewhat to be expected from studies in data domain shift and propensity score models. I would have liked to see more experiments on how these factors affect the bounds, to see how useful the bounds would be in different scenarios.

Overall, the paper is well-written, and the contributions and limitations are clearly described. While the theoretical perspective for fairness estimation with missing data is novel, I am leaning towards reject due to concerns about practical impact, as mentioned above.

Minor comments:
- I found the terms “complete case domain” and “complete data domain” to be not very distinctive and confusing at times. However, I do understand these are terms often used in the literature.
- It seems the estimation approach here can be applied to other domain shifts such as under-representation of samples for certain groups. More explicit and detailed discussion on this would be interesting.
- Why was the upper bound assessed only on classification tasks and the lower bound only on regression tasks?
- Figure 2 caption: typo (b) -> (c)

**Time Spent Reviewing:**

4

---

> ### Author Response · Authors · 2021-08-10
> **Response to Reviewer 3**
>
> Thank you for your very thoughtful and constructive comments! We would like to provide point-by-point responses to your comments below:
>
> **R3-C1: Generalization to other fairness notions**
>
> We agree that it would be clearer to include a detailed discussion about our statement on generalization to other fairness notions in the paper. It was omitted due to the page limit. The specific form of $\mathcal{E}_a$ (page 4, line 136) corresponds to a specific choice of fairness notion and determines the fairness estimand $\Delta_T(g)$. The framework of our proofs can be adapted to other forms of $\mathcal{E}_a$ such as the $L_p$ loss. Roughly speaking, in the case of $L_p$ loss with a general $p$, the specific form of the covering number $\mathcal{N}$ and that of the variance term Var($f(Z)$) need to be changed in the analysis of the upper bound; the specific form of the variance term $\sigma$ needs to be changed in the analysis of the lower bound. We will include this discussion in the updated version, provided additional space is allowed.
>
> **R3-C2: Practical impact of the theoretical results**
>
> The theoretical bounds can provide insights about convergence rate of the fairness estimator and factors that may impact fairness estimation in practice.
> Regarding the upper bound in Equation (1), it provides information about the convergence rates. The first term in the upper bound indeed requires knowledge about the total variation distance, which is not available in practice. However, we would like to highlight that, when the weight estimates are consistent (unbiased), the first term (i.e., the total variation distance term) in the upper bound will be of root-n order and the 2nd term will then become the dominant term. This suggests that, when sample size is large and the weights are estimated consistently in real world applications, the second term in the upper bound, which can be approximated using observed data, can be a good approximation to the upper bound. In addition, the first term in the upper bound would vanish under MCAR so the upper bound can be estimated under MCAR.
> Regarding the lower bound, it is established under mild conditions on the data distribution. As stated in Remark 5, if one is willing to make additional assumptions on the data distribution, e.g., tail behavior (gaussian or sub-Gaussian), it is possible to establish a lower bound with a higher probability. In addition, as discussed in Remark 6 in Section 3, similar lower bounds (i.e., ones held with low probability) exist in the domain adaptation literature.
>
> **R3-C3: Additional experiment on how factors such as different PS models affect the bounds**
>
> We agree that this is a very interesting question to investigate. We will include  additional related experiment results in the updated version, provided that additional space is allowed.
>
> **R3-C4: Connections with other domain shifts problems**
>
> We agree that our approach can be adapted and applied to other domain shift settings such as in the presence of selection bias where certain groups are under-represented due to biased sampling. If space permits, we will include a related discussion in the updated version.
>
> **R3-C5: “The upper bound assessed only on classification tasks and the lower bound only on regression tasks”**
>
> To clarify, in Figure 2-a, we examine both the upper and lower bounds on a regression problem. We will also provide assessment of the lower bound in classification problems, if additional space is allowed.
>
> **R3-C6: “Figure 2 caption: typo (b) -> (c)”**
>
> Thanks for pointing out the typo. It should be (c) instead of (b).

---

### Official Review · Reviewer_Y7HD · 2021-07-16

**Rating:** 7
**Confidence:** 3

**Summary:**

In this paper the authors tackle the problem of ensuring fairness between subgroups in the presence of missing data, as the missingness can be biased between classes. The authors propose upper bound as well as lower bounds on the generalization error when propensity weights have to be estimated. The former is novel but classical, the latter is novel but also greatly original.

**Limitations And Societal Impact:**

Yes the authors have adequately addressed the limitations and potential negative societal impact of their work.

**Main Review:**

The authors place themselves in the setting where some data can be censored at random and fairness between classes needs to be enforced.
The authors correctly give a parallel to the domain adaptation setting but the same setting can be found in the survival analysis literature where similar upper (not lower) bounds exist [1]

[1]G. Ausset, S. Clémençon, and F. Portier, “Empirical Risk Minimization under Random Censorship: Theory and Practice,” Available: http://arxiv.org/abs/1906.01908

The main result, in Theorem 1, provides an upper bound of the disparity in fairness while Theorem 2 provides a lower bound. Both results are theoretically very interesting and the proofs bring valuable methods.

I have, however, some remarks about the experiments:
1/ First the results for the true weights should not be in bold as those performances are impossible to achieve.
2/ Following 1/, this means that in 3 out of 6 cases the best estimator is the unweighted one. This needs to be mentioned and analysed. Why is that?
3/ Why do the SVM w perform so badly? Any insights?

Other than those small concerns and the review of literature that needs to be improved, this paper succinctly studies and solve a very interesting theoretical problem. I think it is a great fit for NeurIPS. I do not have any additional remarks, but given the limited time to review the proofs I can have missed errors.

**Time Spent Reviewing:**

4

---

> ### Author Response · Authors · 2021-08-09
> **Response to Reviewer 2**
>
> Thank you for your very thoughtful and constructive comments! We would like to provide point-by-point responses to your comments below:
>
> **R2-C1: Missed relevant work [Ausset et al. 2019]**
>
> We will include the paper [Ausset et al. 2019] in our reference in the updated version. While there is a connection between censoring and missing data, the key difference between their work and our work is that their work did not investigate the fairness estimation in the presence of censoring data. While the form of the upper bound in the main result (Theorem 4) in [Ausset et al. 2019] looks similar to the upper bound in our work, theirs is established for the learning rate of the minimizer of the so-called Kaplan-Meier risk, while ours is for the bias in fairness estimation. In addition, their work does not investigate lower bounds.
>
> **R2-C2: Experiments**
>
> *C2.1: true weights in bold*
>
> We will remove the bold font for the results for the true weights in the updated version. The results for the true weights are used as a benchmark to help assess the performance of the other weights in terms of whether their bias in fairness estimation is close to this benchmark.
>
> *C2.2: outperformance of unweighted estimator*
>
> According to Table 1, out of the total 6 settings, the unweighted estimator has the best (averaged) performance in two settings: MCAR in the COMPAS dataset and MNAR in the ADNI dataset. We would like to clarify that, under MCAR mechanisms, the unweighted fairness estimator is also consistent. Hence it may yield the best performance in some cases under MCAR. Under MNAR, all the estimated weights are mis-specified. In such cases, the relative performance among different estimators may depend on the specific setting and may vary from one setting to another.
>
> *C2.3: performance of SVM*
>
> We think that the under-performance of SVM may be related to the specific data distribution in the COMPAS dataset, which is likely a pathological case. Of note, SVM yields good performance in the ADNI dataset.

---

> > ### Comment · Reviewer_Y7HD · 2021-08-30
> > **Thanks**
> >
> > Thank you for the clarifications.
> >
> > I agree with the other reviewers that applicability of the results seem lacking and therefore deserves a more in-depth treatment. However I still think that those results are useful even in a vacuum and therefore make an interesting paper.
> >
> > Given this I have decided to keep my score but lower my confidence (for that score) to better take into account the remarks from the other reviewers.

---

### Official Review · Reviewer_RPBw · 2021-07-19

**Rating:** 7
**Confidence:** 3

**Summary:**

This paper discusses how to estimate fairness measures in the presence of missing values. Especially, it compares the fairness measure for the complete data domain (data distribution assuming no missing values) and the fairness measure estimated with reweighing on the complete case domain (data distribution after dropping all the data points that have missing values). They provide an upper bound and a lower bound on the difference of the two. They show experimental results of estimating the fairness measures on synthetic data and two real-world datasets to evaluate.

**Limitations And Societal Impact:**

I did not find any statement regarding this in the paper.

**Main Review:**

While fairness in machine learning and handling missing values in the machine learning pipeline have been active areas of research, the intersection of them — ensuring/assessing fairness in the presence of missing values — is a relatively new and unexplored area. Existing works have mostly focused on empirical studies and there has been little to no theoretical study on the topic.

This paper provides a theoretical study on estimating group fairness metrics, such as accuracy parity gap, in the complete case domain. This problem can be thought of as statistical estimation under a domain shift, and hence share the similarity with the results in [37], which studies fairness in transfer learning. However, transfer learning setting specifically for the missing values problem is new. Also proof-technique-wise, the paper claims to make some advances from the existing works in domain adaptation, but I did not check the proofs.

As the authors acknowledge, the paper considers only the simplest setup where we drop a row if it has any missing values, which might not be the most practical scenario. Furthermore, the paper only evaluates the bias on two generic datasets which do not have missing values and they had to generate artificial missing values. The lack of real-world evidence and the simple problem setup limits the practical value of this work. However, I believe that it is an important first step to establishing a theoretical understanding of fairness in the presence of missing values.

The paper is very well-written and clear.

**Time Spent Reviewing:**

8

---

> ### Author Response · Authors · 2021-08-09
> **Response to Reviewer 1**
>
> Thank you for your very thoughtful and constructive comments! We would like to provide point-by-point responses to your comments below:
>
> **R1-C1: Simple setup (complete case analysis) which might not be practical**
>
> Complete case analysis is actually often used in practice, particularly in biomedical studies, and has not been investigated in the terms of fairness, so it is valuable to study this set-up. In addition, this work lays a foundation for more sophisticated analysis of fairness in the presence of missing data, as stated in the discussion section.
>
> **R1-C2: Lack of real-world evidence (analysis of dataset with actual missing values)**
>
> While we can obtain fairness estimators (in the complete case domain) on real datasets with actual missing values, we would not be able to assess the bias in fairness estimation since the true fairness of an algorithm (in the complete data domain) would not be available. In addition, we would not be able to use the true propensity score model for missingness as a valuable benchmark. As a result, we chose to generate artificial missing values in real datasets, which allows us to compute the bias in fairness estimation and use the true propensity score model as a benchmark while making the experiments more realistic.

---

### Decision · Program_Chairs · 2021-09-27

**Decision:**

Accept (Poster)

**Comment:**

The reviewers agreed that this is a well-written paper that addresses a relatively unexplored facet of fair ML: the impact of missing data on fairness. The setup considered in the paper is simple, yet amenable towards an interesting theoretical analysis. However, the paper also has limitations, particularly in terms of how the bounds can translate to "fair imputation methods," as noted by reviewer ykYW. I agree with this concern, and wished that the authors had gone further than the results in the paper and discussed methods to *ensure* fairness in the presence of missing data. The reviewers also noted that the experimental results are on datasets where entries are artificially missing, raising concerns about the practical impact of the paper in setting where data missingness may be correlated with group attributes.

I also found the claim "Our work provides the first known results on fairness guarantee in analysis of incomplete data" to be a bit misleading, since there have been a few papers that at least discuss the topic (e.g., "Missing the missing values: The ugly duckling of fairness in machine learning" by Fernando et. al and https://arxiv.org/abs/1911.12587), albeit to a less theoretical extent.

Overall, the merits of this paper outweigh is limitations, and its publication will encourage more discussions on the impact of missing data on fairness.